# Turing patterns with high-resolution formed without chemical reaction in thin-film solution of organic semiconductors

Zezhong Xiang[1,5], Jin Li[2,5], Peng You[1], Linbo Han[3], Mingxia Qiu[1], Gengliang Chen[4], Yu He[1], Songqiang Liang[1], Boyuan Xiang[1], Yaorong Su[1], Hongyu An[1] & Shunpu Li [1][✉]

Regular patterns can form spontaneously in chemical reaction-diffusion systems under non-equilibrium conditions as proposed by Alan Turing. Here, we found that regular patterns can be generated in uphill-diffusion solution systems without a chemical reaction process through both in-situ and ex-situ observations. Organic semiconductor solution is confined between two parallel plates with controlled micron/submicron-meter distance to minimize convection of the liquid and avoid spinodal precipitation at equilibrium. The solvent evaporation concentrates the solution gradually into an oversaturated non-equilibrium condition, under which a phase-transition occurs and ordered concentration-waves are generated. By proper tuning of the experimental parameter, multiple regular patterns with micro/nano-meter scaled features (line, square-grid, zig-zag, and fence-like patterns etc.) were observed. We explain the observed phenomenon as Turing-pattern generation resulted from uphill-diffusion and solution oversaturation. The generated patterns in the solutions can be condensed onto substrates to form structured micro/nano-materials. We have fabricated organic semiconductor devices with such patterned materials to demonstrate the potential applications. Our observation may serve as a milestone in the progress towards a fundamental understanding of pattern formation in nature, like in biosystem, and pave a new avenue in developing self-assembling techniques of micro/nano structured materials.

Patterns occur in nature at various scales ranging from small atomic level to large scale of the universe[1,2]. At the conventional scale we see snow-flakes, ladybird's spots, zebra stripes, sand ripples etc. From a thermodynamic point of view, the patterns in nature could be classified into two categories. One is formed under thermodynamic equilibrium, such as crystal structures of major maters. While, another type is that formed under none-equilibrium condition, such as surface waves of water, animal's skin patterns. For the former case, the pattern/structure formation mechanism has been well studied which is controlled by a minimization of Gibbs free energy[3,4]. For the second case, equilibrium thermodynamic principle is no longer applicable, and one need to discuss the pattern formation from a nonequilibrium

[1]College of New Materials and New Energies, Shenzhen Technology University, Shenzhen 518118, China. [2]Electrical Engineering Division, University of Cambridge, 9 JJ Thomson Avenue, Cambridge CB3 0FA, UK. [3]College of Health Science and Environmental Engineering, Shenzhen Technology University, Shenzhen 518118, China. [4]Sino-German College of Intelligent Manufacturing, Shenzhen Technology University, Shenzhen 518118, China. [5]These authors contributed equally: Zezhong Xiang, Jin Li. ✉e-mail: lishunpu@sztu.edu.cn

**Fig. 1 | Schematic illustration of proposed concentration-wave formation process. a** Solution film sandwiched between a PDMS plate and the substrate. **b** Proposed concentration-wave generated in solution where the color contrast in the top panel corresponds to the solute distribution in the liquid, which is further illustrated by a wavy curve (bottom panel). **c** Schematic illustration of a line-pattern formed on the substrate through deposition of the generated waves. **d** Schematic illustration of molecule-solvent binary phase diagram where spinodal and binodal curves are displayed. The red dashed curve represents the modified spinodal curve after PS addition. The inset shows a schematic spinodal structure formed under equilibrium.

thermodynamic point of view[5–7]. In 1952, Alan Turing suggested a possible connection between the patterns in biological systems and patterns that could form spontaneously in chemical reaction-diffusion systems[8]. At the time, there were several important experimental facts to support the theory, for instance Bray oscillation and Belousov-Zhabotinsky reaction, and proved the existence of spontaneously spatially or temporally modulated chemicals (i.e. chemical waves)[9–11]. In 1970s, based on extensive researches in this field, Pringogine et al. concluded that these chemical waves/oscillations were stable spatio-temporal structures, named as "Dissipative Structure", which are formed under conditions far away from equilibrium[12]. They concluded that a system needs to meet the following conditions for emerging of the ordered dissipative structures: (i) The system is open in a state far from equilibrium with nonlinear dynamic process, and (ii) the pattern formation is a self-catalysis process with positive feedback.

Among dissipative systems observed, Turing's reaction-diffusion and Rayleigh-Bénard (R-B) convection are the two representative systems for spatial pattern formation, and they have been actively studied because of their analytical and experimental accessibility[7,13–24]. Although the appeared structures either in scales or formation process are different for diverse dissipative systems, their morphologies are similar[25]. For example, line patterns, square-grid arrays, zig-zag patterns etc. have been observed in both Turing and R-B systems[25–27]. This unusual feature can help us to use a "comparative study method" to identify the pattern formation mechanism in unfamiliar systems.

In this work, we found that Turing patterns can also be generated in an uphill-diffusion system without a chemical reaction process through in-situ and ex-situ observation. In oversaturated thin-film solutions of organic semiconductors, regular concentration-waves have been generated and wave-property (like wave amplitude growth and phase velocity etc.) was characterized. Various patterns with micro/nano-meter scaled features have been observed, including line, square-grid, zig-zag, fence-like patterns etc., which are analogous to that found previously in Turing and R-B systems. In addition, we have fabricated and characterized field effect transistors (FETs) by using the

self-assembled structures to demonstrate the potential of the formed patterns.

## Results and Discussion
### Pattern formation experiments

Small molecule organic semiconductor 2,7-dioctyl[1]benzothieno[3,2-b][1] benzothiophene (C8-BTBT) has been chosen for this research. Solutions with various concentrations of C8-BTBT in chlorobenzene (2-20 mg/ml) were made at ambient conditions. The mixture of C8-BTBT and polystyrene (PS, Mn = 10000) with different proportion was used to prepare solutions and investigate the influence of PS addition on the pattern formation. C8-BTBT and PS are well-known organic materials that can be intermixed via solution process[28,29].

Polydimethylsiloxane (PDMS) films with grooved lines for various line-width/separation and groove depth were chosen as cover plates. The surface grooves on PDMS plate are used for solution thickness control only. For ex-situ observation, silicon wafers with oxidized surface layer were used as substrates; while for in-situ observation, glass substrates were chosen. The substrates were cleaned with acetone and isopropanol sequentially and treated with oxygen plasma for 3 min (Gas flow rate: 9.7 sccm; Pressure: 0.5 mbar; Power: 150 W) before experiments. Meanwhile, the PDMS cover plates were treated for various time with other parameters unchanged.

Solution with controlled volume (about 3 μl for PDMS plate with 1 cm² surface area) was drop-casted onto the surface of the PDMS plate. Then, a substrate was gently brought into contact with the solution-wetted PDMS plate (Fig. 1a and Supplementary Fig. 1a) and dried at room temperature for 30 min under an applied pressure with a clamping tool (about 5 Mpa) to generate concentration wave (Fig. 1b). After pattern formation and sample drying, the sample (PDMS plate attached on the patterned substrate) was removed from the clamping tool. Then, the PDMS plate was gently peeled off from the substrate and the patterns remained on the substrate were used for analysis (Fig. 1c). As the surface of PDMS plate is more hydrophobic than the substrate, no pattern damage

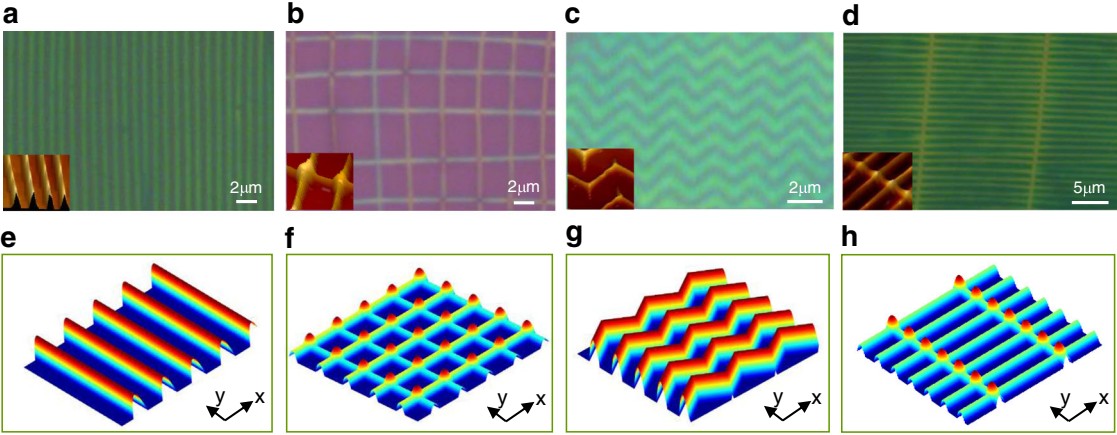

**Fig. 2 | Optical/AFM images (insets) of the self-assembled microstructures and the results of simulations with harmonic waves. a** Line pattern formed with solution (10 mg C8-BTBT + 10 mg PS)/ml. **b** Square-grid pattern formed with solution 10 mg C8-BTBT/ml. **c** Zig-zag pattern formed with solution (10 mg C8-BTBT + 5 mg PS)/ml. **d** Fence-like pattern formed with solution (10 mg C8-BTBT + 10 mg PS)/ml. e Single harmonic wave with vector in the x-direction. **f** Two harmonic waves with vectors in two orthogonal directions. **g** Single harmonic wave of vector in the y-direction with a periodic phase swing. **h** Single harmonic wave of vector in the y-direction superimposes with a wave package of vector in the x-direction. For the simulated patterns only high concentration parts are displayed, as the precipitation starts around the peaks of the waves.

from the PDMS adhesion was observed. Therefore, the PDMS removal process had very minimized impact to the pattern quality and device performance.

The groove width and separation are identical on each PDMS plate and can be changed from 10 μm to 500 μm by using different PDMS plates. The grooves are designed to precisely control the thickness of solution film, and their depth was varying from 1.5 μm to 50 μm. For the majority of experiments the groove depth used was 1.5 μm unless otherwise stated.

## Morphologies of the formed patterns

Figure 2a–d show the patterns on SiO₂/Si substrates formed in solutions with compositions: (10 mg C8-BTBT + 10 mg PS)/ml (a), 10 mg C8-BTBT/ml (b), (10 mg C8-BTBT + 5 mg PS)/ml (c), and (10 mg C8-BTBT + 10 mg PS)/ml (d), respectively. One can see that very regular structures were generated. Depending on the solutions used, array of lines, square-grid, zig-zag, fence-like patterns were generated. The array of lines can be observed in a wide range of C8-BTBT:PS composition ratios (from 1:0 to 1:2). The square-grid pattern is preferred to appear for solutions without PS addition. In contrast, the fence-like patterns can be observed when the solution contains sufficiently large amount of PS. The zig-zag pattern formation rarely happens and can be occasionally observed for solution with controlled composition. The single domain size of the patterns is in the scale of few hundred micrometers which can generate diffraction pattern with the most accessible LED laser beam (Fig. 3a). Small line features (150-300 nm) were obtained with non-optimized experimental condition (Fig. 3b). The line period (λ) for patterns generated with pure C8-BTBT solutions is several micrometers and it reduces to ~1 μm with addition of PS around 30-50% in the solution, and it increases again slightly with further increased PS content. AFM characterization showed that the thicknesses of the patterned structures were in the range 40 nm-90 nm depending on compositions and concentrations of the solutions used for pattern generation. Normally, the crystalline size in the C8-BTBT films can be in the range of sub-micrometer or micrometers[30]. However, no crystalline grains are visible in the AFM images of the Fig. 2. The reason is that the patterned C8-BTBT has nanocrystalline structure with the size of ~20 nm (Supplementary Fig. 2), which might be attributed to the high nucleation density of C8-BTBT under large undercooling.

## Pattern formation mechanism and simulations

A proper surface property of the PDMS plate and appropriate solution layer thickness are required for pattern formation. Regular patterns are formed when the PDMS plate was treated with oxygen plasma for 2-3 min. Longer or shorter time of treatment can deteriorate the pattern formation. The measured water contact angles for the plasma-untreated and treated PDMS film are 109.96° and 10.44°, respectively, displaying dramatically different surface properties (Supplementary Fig. 3). This indicates that the interface energy is a non-negligible factor for pattern formation. To form stable and regular patterns in liquid films convection must be avoided. In our parallel-plate system, the surface tension gradient-induced convection is negligible. Gravity and thermal fluctuation-induced buoyancy convection can be suppressed through the reduction of the distance between the two plates. It was reported that 500μm thick liquid confined in a slit can effectively reduce buoyancy convection[31]. In our experiments, the solute distribution needs to be stabilized in nanoscale regions to fix the patterns in liquid films. This requires micrometer-sized space to prevent the convection strictly which is often used to investigate diffusion in solution[32]. We have fabricated PDMS plates with various groove depths (1.5-50 μm) and found that the patterns are formed when the groove depth is less than ~7 μm. The total liquid film thickness is contributed from the groove depth and a liquid layer trapped between the substrate and protruding parts of the PDMS plate surface (i.e., $a + b$ in Fig. 1a). The trapped liquid film thickness $b$ at the initial experiment was estimated to be 400-600 nm (Supplementary Note 1). The patterns can be formed under both protruding parts (indicated by M in Fig. 1a) and groove parts (indicated by N in Fig. 1a) of the PDMS plate, and better pattern quality in the groove parts was found.

The observed patterns can be classified as Turing patterns because the formation process is diffusion-driven, although no chemical reaction exist. The experimental system satisfies the condition required for dissipative structure formation:

(1) The system is an open system where the solvent evaporates continually, and the solution parameters $P$ change constantly, i.e., $dP/dt \neq 0$. The parameter $P$ can be solution concentration ($C$), solution viscosity ($\eta$), and diffusion coefficient ($D$) etc. Therefore, the system is not in an equilibrium condition, and the variation of diffusion coefficient leads to a nonlinearity of solute transport in the solutions.

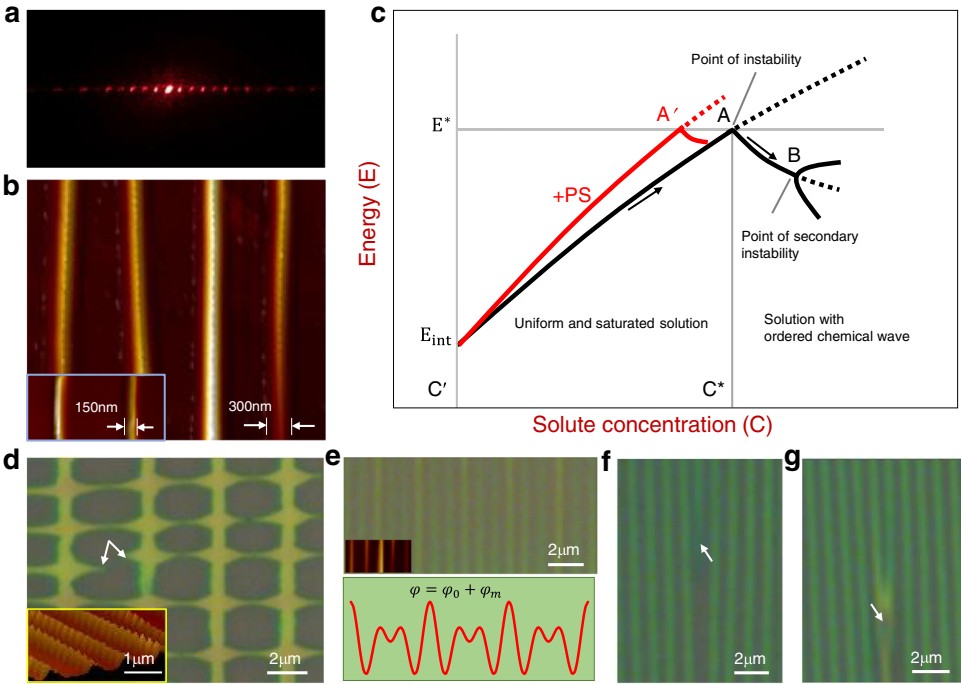

**Fig. 3 | Structure analysis and proposed mechanism for pattern formation.**
**a** Optical diffraction taken from a line pattern with LED diode laser. **b** AFM images of the obtained line patterns which show line features of 300 nm and 150 nm. **c** Proposed pattern formation mechanism which shows schematic energy-concentration relation of oversaturated solution film. $E_{int}$ is the interface energy and $E^*$ is the critical energy density for phase transition to happen. A and B are the first and secondary instability points, and the red colored curve corresponds to the case when PS is added into the solution. **d** Microstructure of an underdeveloped square-grid pattern where a branching from vertical primary lines is seen. White colored arrows indicate misalignment of the branches. The AFM image (inset) shows a modulated feature with unstable lines. **e** Modulated microstructure (top panel) which can be fitted with superposition of two waves (bottom panel). **f** Microstructure which shows 'line-end' dislocation emerged in an array. **g** Microstructure which shows a dislocation fused with neighbor line before passing through it (detailed in Fig. 4b and Supplementary Movie 2).

(2) The uphill-diffusion in the investigated system acts as self-catalysis for the mater transport. In Turing's theory, a self-catalysis of chemical reaction is used to amplify a perturbation and generate regular chemical waves. Here, in the investigated system, the uphill-diffusion controlled spinodal precipitation can have a similar effect. We detail the self-catalysis and concentration wave formation for this work in Supplementary information (Supplementary Fig. 4 and Note 2).

In a given experiment, the Turing pattern generation requires a control factor X which drives the system away from equilibrium. If the system is driven not far away from its equilibrium, it will adopt equilibrium state eventually. However, if the X reaches a critical value $X_C$, the system will no longer adopt its equilibrium state and occupy a new state, i.e., regular Turing pattern is generated. In our experiment, the organic material and solvent form a binary spinodal system[33,34], and its schematic phase diagram is shown in Fig. 1d. The binodal and spinodal curves are defined by $\partial G/\partial C = 0$ and $\partial^2 G/\partial C^2 = 0$, where $G$ and $C$ are Gibbs' free energy and solute concentration, respectively. For a coordinate point (concentration $C$, temperature $T$) within the spinodal region, where $\partial^2 G/\partial C^2 < 0$, a homogeneous solution is unstable against fluctuations in density or concentration and separates to two phases with a well-defined size[35]. This process is generally called spinodal decomposition. In the case of solution system, like an organic solid material dissolved in a liquid solvent, a solid phase is precipitated from the solution during the spinodal decomposition, and such process is also called "spinodal precipitation".

The near equilibrium spinodal structure is random (inset of Fig. 1d) since the weak coupling between different waves. At the beginning of the experiment, the concentration of the clear solution is $C_0$, which is located outside of the spinodal region. With the progressing of solvent evaporation, the solution concentration increases to C in the spinodal region. At concentration C, the non-precipitated solution is oversaturated, which is signified by under-cooling $\Delta T$, and disordered structure might form, once the solution starts precipitate under external disturbance. If the solution-oversaturation can be large enough by minimizing disturbance, the concentration can reach a critical value $C^*$ and the precipitated structures become ordered, i.e., Turing pattern is produced. The order of structures is induced by the coupling between different waves under high undercooling.

The energy density ($E$) stored in the solution is mainly contributed from the oversaturation-induced undercooling ($\Delta T$) and interface energy ($\sigma$) (detailed in Supplementary Fig. 5 and Note 3):

$$E = \beta(C - C')C_p + (\sigma_1 + \sigma_2)/(a + b), \tag{1}$$

Where $C_p$ is the specific heat of the solution which is treated as a constant, $(a + b)$ is the liquid film thickness, $C'$ is the concentration on the spinodal curve corresponding to experimental temperature, $\beta$ is the slope of left branch of the spinodal curve, $\sigma_1$ and $\sigma_2$ is the interface energy density for PDMS/solution and solution/substrate interfaces, respectively. Equation (1) is schematically expressed in Fig. 3c. In our system, the control parameter is solute concentration $C$. Once the solution is condensed to concentration $C^*$, the energy of the system reaches a critical value $E^*$ and a state transition happens (i.e., point of instability, see point A of Fig. 3c). The required moderate oxygen-plasma treatment of PDMS plate could be explained as that, higher interface energy ($E_{int}$, the second term of Eq. (1)) is favorable for the state transition, as the $E$-$C$ curve will shift to higher energy direction, and the critical energy $E^*$ is achieved relatively easier. Meanwhile, a plasma treatment is still required to maintain a reasonable wettability

of the PDMS plate with the solution. The confinement of the liquid to a very thin film is important to drive the system away from equilibrium and to stabilize the generated concentration waves.

The spinodal process of the system is seen from microstructures of drop-casted thin films (Supplementary Fig. 6) and can be further proved through theoretical estimation of spinodal wavelength. The spinodal process defined wavelength $\lambda_{sn}$ is:[35]

$$\lambda_{sn} = 2\pi\sqrt{\frac{4k}{-G''}}, \qquad (2)$$

where $G''$ is the second derivative of the free energy of solution system with respect to composition which can be measured with light scattering or small angle neutron scattering (SANS) techniques[36,37]. The $k$ is the square gradient parameter accounting for change in free energy arising from concentration gradient. It can be accessed by computation or measurement with scattering techniques[38]. By taking $k$-0.05–0.3 ($\text{Å}^2\,\text{mol·cm}^{-3}$) and $-G''$-$10^{-7}$–$10^{-5}$ ($\text{mol·cm}^{-3}$)[38], one can estimate $\lambda_{sn}$ -0.1–3 μm, which agrees with our experiment.

The formation of concentration waves is further supported by both simulation and in-situ observation of pattern generation in the liquid. The simulation results show that all observed patterns can be composed with harmonic waves with proper mode selections (Fig. 2e–h). A generated pattern can be described with $N$ pairs of concentration/chemical waves:[25]

$$C = C_0 + \sum_{i=1}^{N}(A_i \exp(i\mathbf{k_i} \cdot \mathbf{r}) + \bar{A}_i \exp(-i\mathbf{k_i} \cdot \mathbf{r})), \qquad (3)$$

Where $A_i, \bar{A}_i$ are the amplitudes for waves with wavevector $\mathbf{k_i}$ and $-\mathbf{k_i}$, respectively; $\mathbf{r}$ is the position vector. $N=1$ corresponds to the vertical line pattern (Fig. 2a) formed from a wave $\varphi_x = A_x \exp(ik_x x)$ with its wavevector points in x direction (Fig. 2e). Accordingly, $N=2$ corresponds to square-grid pattern formed via interference between two waves with orthogonal vectors (i.e., $\varphi_x + \varphi_y$) (Fig. 2f). The zig-zag structure is simulated utilizing a wave with a periodical phase swing (Fig. 2g). The fence-like pattern is formed through interference between the wave $\varphi_y$ and a wave-package ($\sum \varphi_x^i = \sum A_x^i \exp(ik_x^i x)$) with vector in the $x$-direction (Fig. 2h). The reason why only a limited number of waves are involved in the pattern formation is that a "spatial resonance principle" is applied during the mode selection and only a finite number of waves are stable[25,39]. A distinctive feature of waves is the interference, which occurs when waves with different wavevectors encounter, as clearly seen from the humps at cross points of lines (insets of Fig. 2b, d). For the simulated patterns in Fig. 2e–h, only high concentration parts are displayed for comparison with experiments, because the material precipitation from the solution begins around the peaks of the waves (see later discussion).

Our experiments show that the line pattern can coexist with any one from other three types of patterns in Fig. 2, however, any two from those three cannot stably appear at the same sample. This excludes the possibility of energy degeneracy of the line pattern with other three patterns. Otherwise, those three patterns (or any two of them) should have a chance to coexist. We propose that the line pattern is the preferred state to be adopted once the critical point (i.e., the first instability point, indicated with A in Fig. 3c) is reached, and all other patterns are formed via instability of the line pattern, i.e., newly selected state by the system after line pattern generation (the secondary instability point, indicated with B in Fig. 3c). The speculation is consistent with previous observations in Turing and R-B systems. The square-grid patterns observed here are similar to the cross-line (or cross-roll) patterns generated from the unstable line pattern in the R-B system[26,27]. Such instability is distinguished from some underdeveloped square-grid patterns where one can see that the square-grid patterns are developed through side-branching of lines (Fig. 3d). In

Fig. 3d, the vertical lines are the primary lines (formed from the first instability), which become unstable and side-branches develop from them (the secondary instability). The misaligned sharp features (indicated by white arrows) are the indication of the side growth. In a maturely developed square-grid pattern, the branches from neighbored lines merge completely and form a new set of lines (as shown in Fig. 2b). The collective alignment between the branches is resulted from a predefined modulated diffusion field in response to the newly adopted state. The branching mechanism is required by the energy minimization, since a favorable solute gradient is provided near the primary lines for uphill-diffusion. Germinated line instability is frequently seen as periodically modulated feature on the primary lines (inset of Fig. 3d). Whether these modulated features have chance to grow up depends on the mode selection for adapting to the local environment. We have observed a mode selection process during the square-grid pattern formation: one mode with proper wavelength will become dominant eventually, while, other non-competitive modes will decay (Supplementary Fig. 7). If the initial instability with short wavelength can grow up, and the competition from other long wavelength-mode can be avoided, a fence-like pattern may form. We have also observed that the line instability and branching occur immediately after the primary line emerging. The next neighbored primary line emerges immediately after the branching of the one emerged before it (Supplementary Fig. 8 and Note 4). The zig-zag pattern formation is well known as "zig-zag instability" and has been observed in both Turing and R-B systems. It is originated from phase instability induced by perturbation transverse to the wave vector[40,41], and can be molded by $\varphi = A \exp\{i[k_y y + S(x)]\}$, where $S(x)$ is a periodic sawtooth function. The pattern evolution with increasing PS content is probably related to the modification of spinodal curve of the phase diagram, because the slope $\beta$ enhances when the solute molecule-chain-length increases (red dashed curve in Fig. 1d, and red curve in Fig. 3c)[42]. This leads an increased $\Delta T$ value, and the system can adopt a state which is inaccessible without the addition of PS. The effect of introduction of PS can be understood better with ternary phase diagram (Supplementary Fig. 9 and Note 5). Another instability of the line patterns we have observed is Eckhaus instability, which has also been found in Turing and R-B systems, where a fundamental wave is modulated by another wave[40,43]. Figure 3e shows a pattern formed with such combined wave (top panel) which can be fitted with superposition of a fundamental wave $\varphi_x = A \exp(ik_x x)$ and a modulation wave $\varphi_m = A \exp(i2k_x x/3)$ (bottom panel). This is interpreted by a local environment fluctuation (like concentration) that readjusts the wavelength. A common defect of the line pattern is 'line-end' dislocation (Fig. 3f, referred as dislocation hereafter). The dislocation can migrate in the array of lines to adjust the line separation, or to improve the integrity of the array by migrating to domain boundary. Figure 3g shows an intermediate stage of the migration, where the dislocation is fused with a neighbor line that the dislocation attempts to pass through (further detailed in Fig. 4b). The pattern formation can be generalized in other systems if the proposed conditions are met. We have tried other systems other than C8-BTBT(PS) solutions. Pattern formation has also been observed in certain systems, although the area and regularity of the formed patterns are not ideal. For instance, line pattern of perylene was formed with solution of perylene in chlorobenzene (10 mg/ml), and modulated line pattern or mosaic-like pattern of poly(3-hexylthiophene) (P3HT) were formed with solution of the P3HT in chlorobenzene (10 mg/ml). AFM and X-ray analysis of these patterns is detailed in Supplementary Note 6, Supplementary Figs. 10, 11.

Besides, since our investigated system is liquid-film based, in order to confirm that the patterns are not originated from R-B convection, we have performed experiments with the sample-clamping-tool upside-down, and as a result, the same assembled patterns were obtained, which is impossible to happen for the R-B system (Supplementary Fig. 12 and Note 7).

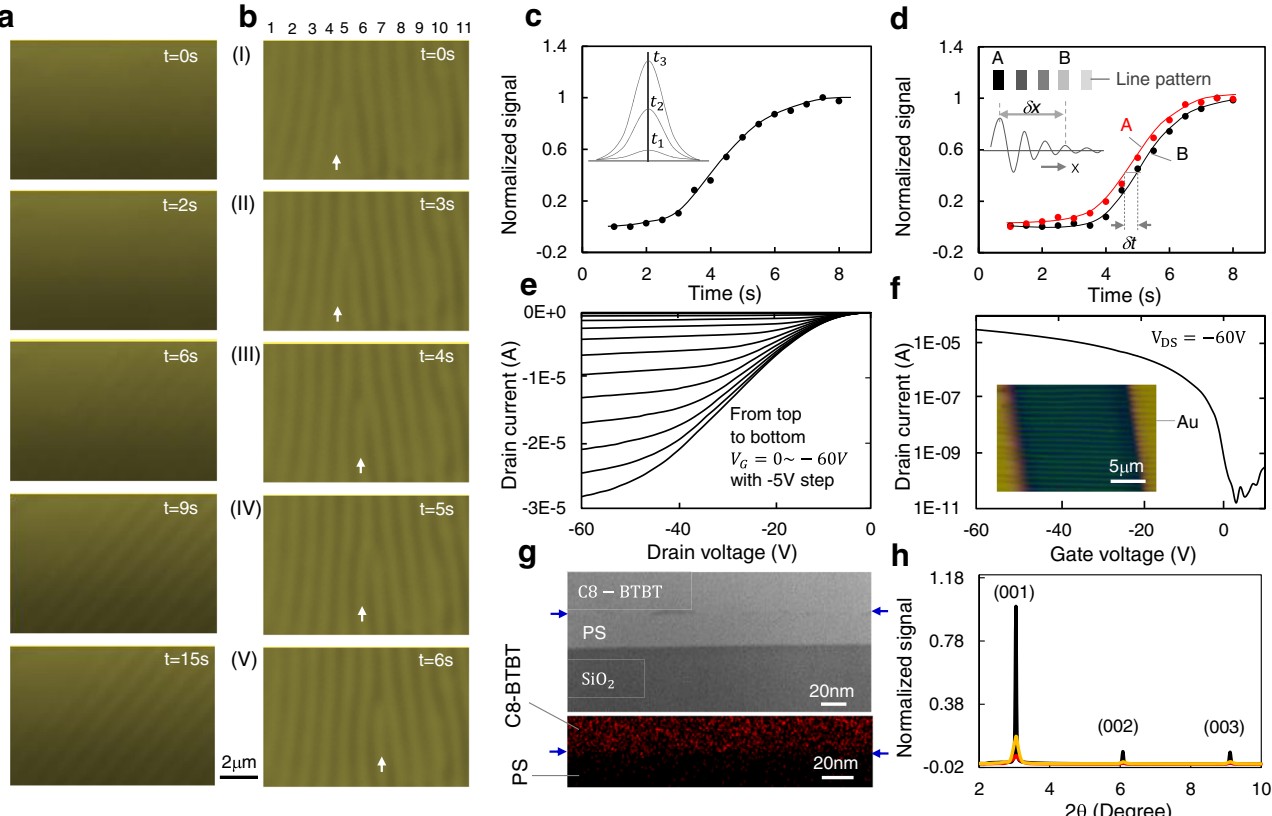

**Fig. 4 | In-situ observations of the pattern formation processes and film/device characterizations. a**, **b** Selected frames in movies taken from in-situ observation, where (a) shows a fast wave-generation and (b) is pattern evolution process which shows that a line-end dislocation has sequentially passed through two neighbor lines on the right. **c** Normalized brightness-contrast between neighbored peak and valley measured at various times during wave formation. **d** Similar measurement as in (c) on two lines with separation $3\lambda$ (indicated with A, B) in order to find phase delay time $\delta t$. The inset illustrates the wave propagation scenario during the wave generation. **e**, **f** Output and transfer curves taken from a FET device made with C8- BTBT + PS wires, where absolute value of the drain current was used for log-plot of the transfer curve. Inset is an image of a device with C8-BTBT + PS wires. **g** The cross-sectional TEM image of a sample of C8-BTBT + PS wires (top panel) and the EDX analysis of sulfur element (bottom panel). The interface between the C8-BTBT and PS layers is indicated with blue arrows. **h** X-ray diffraction spectra taken from patterned C8-BTBT + PS line structure before annealing (black curve) and spin-coated C8-BTBT + PS film before (red curve) and after (yellow curve) thermal annealing.

## In-situ observation of the pattern-formation

More evidences of the concentration waves were obtained through in-situ observation of pattern-formation in solution of (10 mg C8-BTBT + 20 mg PS)/ml in chlorobenzene (Supplementary Fig. 1b). Figure 4a displays selected frames from a video which show a fast wave-generation process (see Supplementary Movie 1). Figure 4b shows the pattern evolution process which reveals that a 'line-end' dislocation initially located between line-4 and line-5 (white arrow in Fig. 4b-I) has passed through line-5 and line-6 (Fig. 4b-(II-V)) sequentially (see Supplementary Movie 2). The migration of the dislocation in Fig. 4b-I starts from fusing with line-5 to form a branched-line (Fig. 4b-II). Then, the dislocation quickly replaces the bottom-half of the line-5 through chopping it off to form a new dislocation (Fig. 4b-III). The same process is repeated for the dislocation to pass the line-6 (Fig. 4b-IV, Fig. 4b-V). The concentration wave can develop-up in about 4-5 s, which was estimated by measurement of time dependence of brightness contrast between neighbored peak and valley in a movie (Fig. 4c). By fitting the curve shown in Fig. 4c we can express the relation between the concentration at wave-peak ($C_{wp}$) and time (t) as an error function:

$$C_{wp} = C^* + \frac{A_m}{2}\left\{ erf[\xi(t - t')] + 1 \right\} \qquad (4)$$

where $C^*$ and $A_m$ are the critical concentration of the solution at the instability point and the wave amplitude, respectively. The moment at

which the time starts to account can be chosen arbitrarily, and t' is the time corresponding to half-development of the wave. The $\xi$ is a constant which can be chosen as $\xi = 0.56$ in our experiment. The rationality of the Eq. (4) can be checked with a simple discussion. Before development of the concentration wave ($t \ll t'$), from Eq. (4) we have $C_{wp} \to C^*$; while, after completion of the wave development (t ≫ t'), we have $C_{wp} \to C^* + A_m$; if we take $t = t'$, then $C_{wp} = C^* + A_m/2$. This is in line with our expectation.

One can obtain phase velocity of the generated wave by doing the same measurements as in Fig. 4c performed on two points (A, B) with distance $\delta x$ during the pattern development (Fig. 4d). During the wave propagation, the wave development of the two points has a time delay $\delta t$, and the phase velocity:

$$v_p = \delta x / \delta t \qquad (5)$$

In our experiment, $\delta x = 3\lambda$, $\delta t$ is taken from Fig. 4d, and we have $v_p$-8.8$\lambda$/s (-12 μm/s). Here, the wave propagation only occurs at the stage of wave generation. Once a wave is fully developed, it becomes stable (see Supplementary Note 2).

The condensation of the concentration waves from the solution onto the substrates provides a useful way for ex-situ study of the pattern formation, which also offers a new material patterning method. We propose a two-step mechanism of the pattern condensation: (i) solute precipitation at the peak regions of the formed

waves and (ii) patterning of residual liquid by the precipitated organic material (detailed in Supplementary Note 8 and Supplementary Fig. 13–15). During deposition of the concentration waves onto the substrate, the solute molecules tend to accumulate to the early-stage deposited structures (like center part of a line). This results clean spaces between the deposited materials (i.e., zero thickness). To better understand the pattern formation, a proposed solvent evaporation and solution concentrating process in the experiments is illustrated in Supplementary Fig. 16 and described in Supplementary Note 9.

### Organic semiconductor devices with the patterned materials

The high regularity and high resolution of the generated patterns have motivated us to fabricate organic nanowire devices, like field effect transistors (FETs), to demonstrate the potential of the patterned materials. Such electronic devices based on semiconductor wires are interesting for various applications, like brain mapping and synaptic devices with low energy consumption[44–46]. We have first generated C8-BTBT + PS line patterns on 300-nm-thick SiO$_2$ films grown on highly doped silicon substrates. Then, the samples were annealed at 60 °C for 4 h in nitrogen-filled glovebox, and subsequently 5 nm MoO$_3$ and 50 nm gold source/drain electrodes were thermally evaporated through shadow mask to complete the FETs fabrication. Figure 4e, f show the output and transfer curves of a fabricated device. The on/off current ratio of the FET device is ~10$^6$, and the derived charge mobility is 2.5 cm$^2 \cdot$V$^{-1} \cdot$S$^{-1}$. The charge mobility was calculated using the expression in ref. 47. with a small modification, that is, deduction of the interline space area by replacing channel width $W$ with $\alpha W \lambda^{-1}$, to suit the nature of our devices. Where, $\alpha$ is the width of the semiconductor lines and $W$ is the channel width of the device. The capacitance of unit area $C_O$ is calculated from the formula $C_0 = \varepsilon_0 \varepsilon_{ps} \varepsilon_{ox} / (\varepsilon_{ps} d_{ps} + \varepsilon_{ox} d_{ox})$, where $\varepsilon_{ps}$ and $\varepsilon_{ox}$ is the relative permittivity of PS and SiO$_2$ layer, respectively; $d_{ps}$ and $d_{ox}$ is the thickness of PS and SiO$_2$ layer; $\varepsilon_0$ is vacuum permittivity. Taking $\varepsilon_{ps}$ ~ 2.5, $\varepsilon_{ox}$ ~ 3.9, $d_{ps}$ ~ 30 nm, $d_{ox}$ ~ 300 nm, $\varepsilon_O$ ~ 8.854 × 10$^{-12}$ F/m, we have $C_O$ ~ 6.93 × 10$^{-5}$ (F/m$^2$). The layer of PS is formed from vertical phase separation occurred in the wires. Specifically, the PS layer is emerged between the C8-BTBT and SiO$_2$ caused by the difference in the surface energies[48,49]. As the methyl-terminated C8-BTBT has lower surface energy than either substrate or PS surfaces, the energy is minimized when PS segregates to the substrate and C8-BTBT surface is exposed[28].

The obtained charge mobility is in the range of 1.7-2.5 cm$^2 \cdot$V$^{-1} \cdot$S$^{-1}$, on/off current ratio varies between 1.4 × 10$^6$ and 7.5 × 10$^6$, threshold voltage varies between 0 V and +3 V. The obtained charge mobility is still relatively lower than many reported results made from continuous films[28,29], and this is caused by low crystal quality of the semiconductor and contact resistance with electrodes. Although a MoO$_3$ layer is introduced to reduce the contact resistance, the charge injection can still be improved which is proved from the sublinear characteristic of the output curves at low drain voltage (Fig. 4e). The moderate hysteresis of the transfer curves (Supplementary Fig. 17a), obtained with forward and reverse gate-voltage scans, suggests a high quality of PS layer which screens trapping effect from the SiO$_2$, because the hysteresis reflects the charge trapping process[50].

Figure 4g displays cross-sectional transmission electron microscopy (TEM) image of a fabricated C8-BTBT + PS line (top panel) and energy dispersive X-ray spectroscopy (EDX) analysis to show the phase separation (bottom panel). As the C8-BTBT molecules contain sulfur element (Supplementary Fig. 17b), the sulfur distribution analysis shows clear double-layer structure resulted from the phase separation. We have carried out X-ray diffraction on both patterned structures and spin-coated continuous films of C8-BTBT + PS (2:1) mixture before and after thermal annealing for comparison (Fig. 4h). The result showed that non-annealed continuous film (~40 nm) contains very little crystalline phase as indicated by the weak diffraction peaks (red curve). After thermal annealing of the continuous film at 100 °C for 4 h the

diffraction peaks became more obvious as expected which is originated from crystallization of C8-BTBT from the mixture (yellow curve). In contrast, for the patterned structures without thermal annealing the diffraction strength is much stronger (black curve) than that taken from the spin-coated continuous films, although the material covered by the X-ray beam for the patterned sample is not more than that for the continuous films. This can be explained as that the patterned structures were in solvent environment for much longer time (~minutes) than the case of spin-coated films (~seconds). This allows the C8-BTBT molecules to aggregate and crystalize. No obvious change of the diffraction strength was found after thermal annealing the patterned C8-BTBT + PS structures at 100 °C for 4 h. We have also carried out X-ray analysis on samples with patterned structures of pure C8-BTBT before and after thermal annealing, and very similar diffraction spectra were obtained (Supplementary Fig. 18).

In this work we used structured PDMS plates (stamps) to confine solution and investigate pattern formation of organic semiconductors (OSC). Comparing with other works for OSC patterning using stamp techniques (Supplementary Table 1) this work has its own distinctiveness: (i) In this work, the structure on a PDMS stamp is used to control solution thickness only, and the lateral dimension of the structure is not important. In contrast, for the methods listed in the table the lateral dimension of structures on stamps is crucial to define the size and geometry of patterned OSC. (ii) The patterns formed in this work are self-organized, instead of artificially designed. (iii) In this work, the patterns are formed under non-equilibrium condition, which is different from those where the patterned materials are formed under (or close to) equilibrium condition. On the other hand, this work also shares some common features with many other works, like surface energy tuning of the stamps/substrates, flexibility control of stamps for firm contact during printing etc. With further development this work may have potential to fabricate nanomaterials with low cost. The challenge is that the variety of formed patterns are limited, and the size of patterned structures needs to be controlled very carefully.

In summary, we have observed Turing pattern formation in geometrically confined oversaturated solution films of organic semiconductors without chemical reaction process. By stably concentrating the confined solution films, highly ordered patterns including line, square-grid, fence-like, and zig-zag etc. patterns can be generated. These pattern morphologies and pattern instability are typical features of pattern-formation in the well-known dissipative systems, like Turing's reaction-diffusion and Rayleigh-Bénard convection systems. The investigated system satisfies the required condition for being a dissipative system, particularly the uphill-diffusion can form a self-catalysis process. To observe ordered structures, the system needs to be driven far away from equilibrium without precipitation before the instability point (phase transition point) is reached. In a thin solution film sandwiched between two solid surfaces, the convection can be suppressed which is beneficial to achieve a stable nonequilibrium state and generate/stabilize the concentration waves. The patterns observed here can be classified as Turing patterns since the wave formation is diffusion-driven. The experimental system we used is so simple that such patterns could be generated in natural conditions without artificial involvement. We are not clear whether it has connection with certain patterns formed in nature (like in biosystem) at present, which requires more investigation. With further development, the generated patterns might have potential applications as nanomaterials.

## Methods

The structured PDMS plates were duplicated by pouring commercial silicone elastomer (Sylgard®184, Dow Corning), supplied as a two-part liquid component kit, with a 10:1 mix ratio onto a pre-defined structured masters and annealed at 70 °C for 1 h. For duplication of PDMS plates with groove-depth of 1.5 μm the masters were fabricated with optical lithography, while for the groove depth of 25-50 μm the

masters were made with mechanically cut grid of scotch tapes that firmly appressed on silicon wafers. For the groove depth with 2-20 μm, the masters were made with mechanically carving PMMA films spin-coated on silicon wafers. For the masters with groove depth of 2-50 μm the line width and separation were ~500 μm. Ex-situ experiments were performed on a homemade stainless-steel clamping tool, which provided a proper substrate/PDMS-plate alignment, under the application of controlled pressure (Supplementary Fig. 1a). For in-situ microscopic observation, transmissive mode was used, and samples were made with glass as substrates for light transmission and clamped in a transparent plastic box with open holes for solvent extraction. The pressure for stabilizing PDMS plates on glass substrates is applied by controlling the thickness of PDMS plates clamped in the box (Supplementary Fig. 1b).

The AFM and SEM analyses were performed on Bruker Dimension Icon and FlexSEM 1000 II. TEM experiments were done with JEM-3200FS and FEI Talos200X systems. The X-ray analysis was performed with SmartLab X-ray diffractometer. For FET fabrication, C8-BTBT and C8-BTBT + PS wires were generated on 300-nm-thick thermo-oxidized $SiO_2$ layer carried on highly doped silicon substrates use solution of 10 mg/ml C8-BTBT and (10 mg C8-BTBT + 5 mg PS)/ml in chlorobenzene. After thermal annealing for 4 h at 60 °C, 5 nm $MoO_3$ charge injection layer and 50 nm Au electrodes were thermally deposited through shadow masks. The fabricated devices were measured with Keithley 4200A-SC semiconductor analyzer under nitrogen atmosphere.

## Data availability

The authors declare that the main data supporting the findings of this study are contained within the paper and supplementary information. Source data are provided with this paper. All experimental data are saved on local server at the Shenzhen Technology University and available upon request from the corresponding author via email (lishunpu@sztu.edu.cn). We will do our best to respond to such requests within 14 days. Source data are provided with this paper.

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

## Acknowledgements

The authors would like to thank the support from Guangdong Basic and Applied Basic Research Foundation (No. 2019A1515011673), Education Department of Guangdong Province (No. 2021KCXTD045) and National Natural Science Foundation of China (No. 12274303). P.Y. would like to thank the support from National Natural Science Foundation of China (No. 62104159) and Natural Science Foundation of Top Talent of SZTU (No. GDRC202104).

## Author contributions

Z.X. and J.L. contributed equally to this work. Z.X. has carried out the sample fabrication experiments and sample analysis. J.L. has done simulation and part of theoretical analysis. P.Y., L.H. and G.C. have contributed with AFM analysis. Y.H. has contributed with SEM analysis. SQ. L. helped for device fabrication. B.X. and M.Q. helped with sample fabrication. H.A. and Y.S. have contributed with funding application and manuscript preparation. S.L. supervised/managed the project, carried out experiments, data analysis, and manuscript writing.

## Competing interests

The authors declare no competing interests.
