## [Peer Review File · Nature Communications]

Turing patterns with high-resolution formed without chemical reaction in thin-film solution of organic semiconductorsREVIEWER COMMENTS

Reviewer #1 (Remarks to the Author):

In “Turing patterns with high-resolution formed without chemical reaction in thin-film solution of organic semiconductors”, the authors report a new method for preparing high-resolution regular patterns in uphill-diffusion solution systems, and observe the formation of pattern in-situ and ex-situ observation. The formation of the patterns is explained and simulated. Finally, they demonstrate the potential applications of regular patterns in OFETs. After suitable revision, I recommend publication in Nature Communications.

1. It is well known that C8-BTBT is a highly soluble organic semiconductor, which can be dissolved in most organic solvents at room temperature. The authors said that the regular patterns were formed after removing the PDMS on the plate, while there is no detail about how to remove PDMS, and the removal process whether affects the performance of the devices. In addition, other necessary information also needs to be mentioned, such as the thickness of pattern films.
2. To demonstrate the potential applications, the authors fabricated the OFETs based on the prepared C8-BTBT pattern. However, the OFETs show not ideal device performance, such as quietly low mobility and small on/off current ratio, which is not suitable for preparing advanced organic circuits, and device optimization is necessary.
3. The authors demonstrate that a phase separation between C8-BTBT and PS can be induced by thermal annealing. However, the phase separation may be formed before thermal annealing. In Fig. 4g, they use cross-sectional transmission electron microscopy image and energy dispersive X-ray spectroscopy (EDX) analysis to show the phase separation, while the micrograph can not clearly show a bilayer phase separation with the C8-BTBT on top and the PS layer sandwiched between the C8-BTBT and the substrate.
4. In Fig. 4h, the authors used only one electron diffraction patterns micrograph to show the polycrystalline structure of the C8-BTBT in both the phase-separated wires and the pure C8-BTBT wires is not suitable, and the electron diffraction patterns exhibits poor crystalline quality.

Reviewer #2 (Remarks to the Author):

The authors of “Turing patterns with high-resolution formed without chemical reaction in thin-film solution of organic semiconductors” describe the modelling of various patterns formed in evaporative precipitation of organic semiconductor C8-BTBT using non-equilibrium precipitation arguments. The article shows novel understanding of processes that are created during the formation of OSCs. However, there are several major issues and limitations that makes it unsuitable for publication in this journal.

1. The meaning of “precipitation” is unclear. Although it is expected that C8-BTBT forms a crystalline phase when precipitating out of solution, there are no supporting information such as X-ray patterns that support this. In addition, in these different precipitation scenarios, the impact of nucleation and growth of the films and patterns are not touched upon, which is surprising because AFM and in-situ images are included where crystalline sizes, etc. should be visible. These discussions and data needs to be included to show a full explanation of whether precipitation of the solid phase is the correct phase that is expected.

2. The impact of adding polystyrene to the system needs to be more carefully evaluated. Although it is shown that the addition of PS results in a range of new morphologies, it is unclear what impact the PS is having. The system is first of all a ternary system, so the phase diagrams may (will?) be different for precipitation of C8-BTBT. In addition, no spectroscopic or diffraction data is provided to show what the precipitated material actually is. There is no reason to assume that pure C8BTBT is precipitating out of the solution. In addition, it is unclear whether the patterns that are formed is possible without PS. It would be instructive to show that the patterns obtained can be formed just by controlling diffusion in a binary system, or if this is not possible, state why it is not possible.

3. The model seems to be generalizable to any set of organic molecules as long as the assumptions are met. This claim would be stronger if the same pattern generation were shown with another organic molecule with similar physical properties.

A few other minor points would help with understanding:

4. C^* is described but not shown in Figure 1d.

5. The concept of waves is not defined when it is first mentioned.

6. The role of convection and the limits of the channel height is not explicitly considered. At what heights is the pattern generation expected to disappear?

7. How were the values of k and G'' obtained?

8. A better schematic of how evaporation proceeds in the channel would be instructive to guide the reader to the precipitation process.

9. The impact of PS on transistor performance is unclear. Figure 4g can be better labeled.

10. What are the absolute thickness of the high and low structures?

Reviewer #1 (Remarks to the Author):

1. It is well known that C8-BTBT is a highly soluble organic semiconductor, which can be dissolved in most organic solvents at room temperature. The authors said that the regular patterns were formed after removing the PDMS on the plate, while there is no detail about how to remove PDMS, and the removal process whether affects the performance of the devices. In addition, other necessary information also needs to be mentioned, such as the thickness of pattern films.

Response: After pattern formation and sample drying, the sample (PDMS plate attached on the patterned substrate) was removed from the clamping tool. Then, the PDMS plate was gently peeled off from the substrate and the patterns remained on the substrate were used for analysis. As the surface of PDMS plate is more hydrophobic than the substrate, no pattern damage from the PDMS adhesion was observed. Therefore, the PDMS removal process had very minimized impact to the device performance. AFM characterization showed that the thicknesses of the patterned structures were in the range of 40nm~90nm depending on compositions and concentrations of the solutions used for pattern generation. We have clarified this in the revised manuscript. (See Page 4, Page 5 of Manuscript)

2. To demonstrate the potential applications, the authors fabricated the OFETs based on the prepared C8-BTBT pattern. However, the OFETs show not ideal device performance, such as quietly low mobility and small on/off current ratio, which is not suitable for preparing advanced organic circuits, and device optimization is necessary.

Response: New devices were made and characterized (see the figure below). We purchased new substrates with better quality of SiO₂ layer to minimize gate leakage current of the devices and optimized the thermal annealing parameter of semiconductor. The newly obtained on/off current ratio is 10⁴ which is higher than the previous result for one order of magnitude, and the threshold voltage was reduced for more than 10V (see the Fig. (b, d) below). The newly obtained charge mobility is 1.05 cm²·V⁻¹·S⁻¹ for the devices made with wires of C8-BTBT+PS (see the Fig. (c, d) below). The device performance is still limited by the traps on the SiO₂ substrate even there is a PS layer between the semiconductor and the substrate due to the existed pinholes in the PS layer. Introduction of a hydrophobic SAM (self-assembled monolayer of molecules) layer onto the surface of SiO₂ is a common strategy to reduce the traps. It is difficult to apply this technique in our case, because the pattern generation requires hydrophilic surface of the substrates. Top-gated devices should be a feasible choice to avoid the traps on the substrates, and this demands a development of new gate dielectrics materials and material deposition process. We have described this in the revised manuscript and supporting information. (See Page 13, Page 14, Fig.4e-f of Manuscript, Supplementary Fig.16)

Figure. Output and transfer curves of OFETs made with wires of C8-BTBT (a, b) and wires of C8-BTBT+PS (c, d). The wire patterns were created with solutions of C8-BTBT in chlorobenzene (10mg/ml) and C8-BTBT+PS in chlorobenzene ((10mg+5mg)/ml).

3. The authors demonstrate that a phase separation between C8-BTBT and PS can be induced by thermal annealing. However, the phase separation may be formed before thermal annealing. In Fig. 4g, they use cross-sectional transmission electron microscopy image and energy dispersive X-ray spectroscopy (EDX) analysis to show the phase separation, while the micrograph cannot clearly show a bilayer phase separation with the C8-BTBT on top and the PS layer sandwiched between the C8-BTBT and the substrate.

Response: Yes, the reviewer is correct. The phase separation may happen before thermal annealing. To prove this, we have carried out X-ray diffraction on both patterned materials and spin-coated continuous films of C8-BTBT+PS (2:1) mixture before and after thermal annealing for comparison (see the Fig.(a) below). The result showed that non-annealed continuous film (~40nm) contains very little crystalline phase as indicated by the weak diffraction peaks (red curve). After thermal annealing of the continuous film at 100°C for 4 hours the diffraction peaks became more obvious as expected which is originated from crystallization of C8-BTBT from the mixture (yellow curve). In contrast, for the patterned structure without thermal annealing the diffraction strength is much stronger (black curve) than that taken from the spin-coated continuous films, although the material covered by the X-ray beam for the patterned sample is not more than that for the continuous films. This can be explained as that the patterned structures were in solvent environment for much longer time (~minutes) than the case of spin-coated films (~seconds). This allows the C8-BTBT molecules to aggregate and crystallize in the patterned structures. No obvious change of the diffraction strength was found after thermal

annealing the patterned C8-BTBT+PS material at 100°C for 4 hours. We have also carried out X-ray analysis on samples with patterned structures of pure C8-BTBT before and after thermal annealing, and very similar diffraction spectra were obtained (see the Fig. (b) below). We have described this in the revised manuscript and supporting information. (See Page 14, Fig.4h of Manuscript, Supplementary Fig. 17)

The uncleared phase separation with the TEM micrograph in Fig.4g can be explained by possible surface damage caused by ion-beam cutting during sample preparation and electron beam irradiation during sample analysis. Sample damage is a very common problem for TEM analysis of organic materials. In the revised manuscript we have replaced this micrograph with another one with better contrast to show the phase separation. (See Fig.4g of Manuscript)

Figure. (a) X-ray diffraction spectra taken from patterned C8-BTBT+PS line structure before annealing (black curve) and spin-coated C8-BTBT+PS film before (red curve) and after (yellow curve) thermal annealing. (b) X-ray diffraction spectrum taken from patterned C8-BTBT structure without annealing.

4. In Fig. 4h, the authors used only one electron diffraction patterns micrograph to show the polycrystalline structure of the C8-BTBT in both the phase-separated wires and the pure C8-BTBT wires is not suitable, and the electron diffraction patterns exhibits poor crystalline quality.

Response: Agree with the reviewer that the TEM diffraction quality is poor which is not sufficient for analysis of C8-BTBT polycrystalline structure. We have carried out X-ray diffraction for patterned structures generated from solutions of C8-BTBT (10mg/ml) and C8-BTBT+PS ((10mg+5mg)/ml) in chlorobenzene. Strong diffraction peaks reveal the polycrystalline structures of C8-BTBT in both patterned C8-BTBT and C8-BTBT+PS samples even before annealing. We have replaced the poor electron diffraction pattern with X-ray diffraction spectra in the revised manuscript, and additional information is provided in the supplementary materials. (See Fig.4h of Manuscript, Supplementary Fig.17)

Reviewer #2 (Remarks to the Author):

1. The meaning of "precipitation" is unclear. Although it is expected that C8-BTBT forms a crystalline phase when precipitating out of solution, there are no supporting information such as X-ray patterns that support this. In addition, in these different precipitation scenarios, the impact of nucleation and growth of the films and patterns are not touched upon, which is surprising because AFM and in-situ images are included where crystalline sizes, etc. should be visible. These discussions and data need to be included to show a full explanation of whether precipitation of the solid phase is the correct phase that is expected.

Response: In a binary (or ternary, or system with more components) spinodal system, when a homogeneous phase becomes unstable, it decomposes into two-phase with well-defined structure. This process is generally called spinodal decomposition. In the case of solution system, like an organic solid material dissolved in a liquid solvent, a solid phase is precipitated from the solution during the spinodal decomposition, and such process is also called "spinodal precipitation". This is described in the revised manuscript. (See Page 7 of Manuscript)

Follow the advice of the reviewer, X-ray diffraction was carried out to confirm the formation of crystalline phase of the C8-BTBT. The strong diffraction peaks reveal the crystalline structure of the C8-BTBT in the patterned structures. The X-ray analysis was performed on both patterned material and spin-coated continuous films of C8-BTBT+PS (2:1) mixture before and after thermal annealing for comparison (see the Fig.(a) below). The result showed that non-annealed continuous C8-BTBT+PS film (~40nm) contains very little crystalline phase as indicated by weak diffraction peaks (red curve). After thermal annealing of the continuous film at 100°C for 4 hours the diffraction peaks became more obvious as expected which is originated from crystallization of C8-BTBT from the mixture (yellow curve). In contrast, for the patterned C8-BTBT+PS structure without thermal annealing the diffraction strength (black curve) is much stronger than that taken from the continuous films although the material covered by the X-ray beam for the patterned sample is not more than that for the continuous films. This can be explained as that the patterned structures were in solvent environment for much longer time (~minutes) than the case of spin-coated films (~seconds). No obvious change of the diffraction strength was found after thermal annealing the patterned C8-BTBT+PS materials at 100°C for 4 hours. We have also carried out X-ray analysis on samples with patterned structures of pure C8-BTBT before and after thermal annealing, and very similar diffraction spectra were obtained (see the Fig. (b) below). This is described in the revised manuscript and supporting information. (See Page 14, Fig. 4h of Manuscript, Supplementary Fig.17)

Agree with the reviewer that, normally, the crystalline size in the C8-BTBT films is in the range of sub-micrometer or micrometers [1], which should be visible in the AFM images displayed in the manuscript. However, no crystalline grains were visible in the AFM images of the manuscript. The reason is that the patterned C8-BTBT has nanocrystalline structure with the size of ~20nm (see the Fig.(c) below), which might be attributed to the high nucleation density of solutes under large undercooling. We have displayed the X-ray analysis result and AFM

image in the revised manuscript and supporting information. (See Page 5 of Manuscript, Supplementary Fig.2)

[1] S. Wang, D. Niu, L. Lyu, Y. Huang, X. Wei, C. Wang, H. Xie, Y. Gao. Interface electronic structure and morphology of 2,7-dioctyl[1]benzothieno[3,2-b]benzothiophene (C8-BTBT) on Au film. *Appl. Sur. Sci.* 416, 696-703 (2017).

Figure. (a) X-ray diffraction spectra taken from patterned C8-BTBT+PS line structure before annealing (black curve) and spin-coated C8-BTBT+PS film before (red curve) and after (yellow curve) thermal annealing. (b) X-ray diffraction spectrum taken from patterned pure C8-BTBT structure without annealing. (c) AFM image of C8-BTBT+PS (1:1) line which shows nanocrystalline structure of C8-BTBT.

2. The impact of adding polystyrene to the system needs to be more carefully evaluated. Although it is shown that the addition of PS results in a range of new morphologies, it is unclear what impact the PS is having. The system is first of all a ternary system, so the phase diagrams may (will?) be different for precipitation of C8-BTBT. In addition, no spectroscopic or diffraction data is provided to show what the precipitated material actually is. There is no reason to assume that pure C8BTBT is precipitating out of the solution. In addition, it is unclear whether the patterns that are formed is possible without PS. It would be instructive to show that the patterns obtained can be formed just by controlling diffusion in a binary system, or

if this is not possible, state why it is not possible.

Response: Our experiments showed that the pattern formation can happen in both binary and ternary systems. The addition of third component into a binary system can induce a change of the patterned structures. For instance, in the binary system of C8-BTBT-chlorobenzene square-grid patterns can be found. Line pattern and mosaic pattern can be observed in perylene-chlorobenzene and poly(3-hexylthiophene)-chlorobenzene binary systems (see the response to next comment). We have clarified this in the revised manuscript. (Page 11 of Manuscript, Supplementary Fig. 10)

Agree with the reviewer that it is more reasonable to describe the system use ternary phase diagram to explain the structure change after introduction of PS. We described this in the revised manuscript and supplementary information as below:

Through literature search we can sketch a schematic ternary diagram at experimental temperature as shown with the figure below [1]. The system contains two organic solutes and one solvent (for example: small molecule-polymer-solvent, or polymer-polymer-solvent). We use small molecule and polymer to express the two solutes in the diagram instead of C8-BTBT and PS, and take two points “D” and “E” to describe the effect of polymer addition. The three components of a point in the diagram can be read out from three component-axes of the diagram (three sides of triangle). For instance, one can read out the three components of the point E (small molecule, polymer, solvent) with the length scale of \overline{CQ} , \overline{BP} , and \overline{AM} . We inspect two samples made with prepared solutions at points D and E (hereafter we name the two samples as sample D and sample E, respectively). The two samples contain the same proportion of solvent, and also the same proportion of solute (small molecule + polymer), because both points locate on the line MN. The polymer concentration of the sample E is higher than the sample D. During experiment, with solvent evaporation the ratio of the small-molecule/polymer will not change in the samples before precipitation happens. For the sample D, the concentration will move along the line DF with solvent evaporation. While, for the sample E, the concentration will move along the line EG. We expect that the sample E will reach spinodal curve faster than the sample D, as the solvent contained in point G is more than that in the point F. Therefore, a larger undercooling is expected for the sample E when spinodal precipitation occurs.

For simplicity, we use a quasi-binary expression in the manuscript to show the modified undercooling ΔT with the addition of PS. The description of ternary phase diagram is provided in the supplementary information for readers who are interested in the details.

This is described in the revised manuscript and supplementary information. (Page 10 of Manuscript, Supplementary Note 5, Supplementary Fig. 9)

[1] C. Schaefer, J. J. Michels, P. van der Schoot, Structuring of thin-film polymer mixtures upon solvent evaporation. *Macromolecules* 49, 6858-6870 (2016).

Figure. Schematic illustration of ternary phase diagram of small molecule-polymer-solvent system.

3. The model seems to be generalizable to any set of organic molecules as long as the assumptions are met. This claim would be stronger if the same pattern generation were shown with another organic molecule with similar physical properties.

Response: Agree with the reviewer that the work can be generalized in other systems if the proposed conditions are met. We have tried other systems other than C8-BTBT(PS) solutions. Indeed, pattern formation can be observed in certain other systems, although the area and regularity of the formed patterns are not ideal. Fig.(a) (see below) shows patterned perylene formed with solution of perylene in chlorobenzene (10mg/ml), and Fig.(b) shows a mosaic pattern of poly(3-hexylthiophene) formed with solution of the poly(3-hexylthiophene) in chlorobenzene (10mg/ml). The difficulty of finding very regular patterns with large area in these systems is explained with improper position of spinodal curves and consequently it is hard to drive the system far from equilibrium. We described this in the revised manuscript and supplementary information. (Page 11 of Manuscript, Supplementary Fig. 10)

Figure. Optical images of patterned perylene formed with solution of perylene in chlorobenzene (a) and patterned poly(3-hexylthiophene) formed with solution of the poly(3-hexylthiophene) in chlorobenzene (b).

4. C^* is described but not shown in Figure 1d.

Response: C^* is shown in Fig.1d of the revised manuscript. (See Page 20 of Manuscript)

5. The concept of waves is not defined when it is first mentioned.

Response: We defined the concept of waves in page 2 of the revised manuscript when it is first mentioned as: “..... and proved the existence of spontaneously spatially or temporally modulated chemicals (i.e., chemical waves)”. (See Page 2 of Manuscript)

6. The role of convection and the limits of the channel height is not explicitly considered. At what heights is the pattern generation expected to disappear?

Response: To form stable and regular patterns in liquid films convection must be avoided. In our parallel-plate system, the surface tension gradient induced convection is negligible. Gravity and thermal fluctuation induced buoyancy convection can be suppressed through reduction of the distance between the two plates. It was reported that 500 μm thick liquid confined in a slit can effectively reduce buoyancy convection [1]. In our experiments, the solute distribution needs to be stabilized steadily to fix the patterns in liquid films. This requires micrometer sized space to prevent the convection strictly which is often used to investigate diffusion in solution [2]. We have fabricated PDMS plates with various groove depths (1.5~50 μm) and found that the patterns were formed when the groove depth was less than $\sim 7 \mu\text{m}$. We clarified this in the revised manuscript. (See Page 5 of Manuscript)

[1] T. Ujihara, K. Fujiwara, G. Sazaki, N. Usami, K. Nakajima, Simultaneous in situ measurement of solute and temperature distributions in the alloy solutions. *J. Crystal Growth* 242, 313-320 (2002).

[2] K. Yoshikawa and T Hori, Study on diffusion behavior of dyes in aqueous polymer solution. Part 1: Measurement of diffusion coefficient with improved diaphragm cell method. *SEN'I*

7. How were the values of k and G'' obtained? From reference, more description is needed.

Response: The values of k and G'' were taken from reference [1]. The G'' is the second derivative of the free energy of the mixture with respect to composition which can be measured with light scattering or small angle neutron scattering (SANS) techniques [2,3]. The k is the square gradient parameter accounting for change in free energy arising from concentration gradient. It can be accessed by computation or measurement with scattering techniques [1]. This is addressed in the revised manuscript. (See Page 8 of Manuscript)

[1] J. T. Cabral, J.S. Higgins, Spinodal nanostructures in polymer blends: on the validity of the Cahn-Hilliard length scale prediction. *Prog. Polym. Sci.* 81, 1-21 (2018).

[2] J. T. Cabral and J. S. Higgins, Small angle neutron scattering from the highly interacting polymer mixture TMPC/PSd: no evidence of spatially dependent χ parameter. *Macromolecules* 42, 9528-9536 (2009).

[3] F. S. Bates, P. Wiltzius, Spinodal decomposition of a symmetric critical mixture of deuterated and protonated polymer. *J. Chem. Phys.* 91, 3258-3274 (1989).

8. A better schematic of how evaporation proceeds in the channel would be instructive to guide the reader to the precipitation process.

Response: Agree with the reviewer's suggestion and we use a schematic sketch to illustrate the solvent evaporation and solution concentrating process in the supplementary information as detailed below:

During solvent evaporation the solvent molecules leave from the edges of sample to open air (see the figure below). Let's consider the situation where the mobility of solvent molecules (blue circles) is much larger than that of the solute molecules (black circles) which is reasonable for most cases, because the smaller mass of solvent molecule and the interaction between the substrate and solute molecules [1]. The solvent evaporation starts from sample edges (I, II in Fig.(a)). The top pressure application and capillary effect will force the liquid to fill up the space between the top-plate and bottom-plate. When a solvent molecule leaves from the liquid, a vacancy space is created at the edge of liquid film and it diffuses inward (II, III in Fig.(a)). The movement of vacancy inward equivalents to a movement of solvent molecule outward. The process continues (IV, V in Fig.(a)) and more molecules leave from the sample, and equivalently more solvent molecules move outwards to supplement the vacancy spaces. The vacancies transfer inward and eventually vanish at the liquid/plate interfaces, and the thickness of the liquid film is reduced (VI in Fig.(a)). In this case, the solute concentration in the film is uniformly distributed before precipitation, because the much faster movement of solvent molecules than the solute molecules. In other words, the increased solute concentration at marginal area caused by solvent evaporation can be diluted quickly by the supplied solvent from the internal area. We would like to stress that the molecular movement is not purely controlled by normal diffusion, liquid rheology may play certain role due to the pressure application and capillary force. The rheology does not affect our qualitative discussion, since

the evaporation is a slow and steady process in our experiments. The rheology can result an increased unidirectional mobility of solvent molecules (outward). In contrast, if the mobility of the solvent molecules is not large enough in comparison with the solute, a concentration gradient of solute may build up. This may not greatly affect the pattern formation process, because the pattern formation can occur in a wide range of solute concentrations as shown in our experiments. Fig.(b) is schematic drawing for analysis, where the red line represents a simplified concentration distribution $C(x)$, and $\Delta C(x)$ is the concentration deviation in comparison with the concentration in the center area of sample C_c . While, the black curve represents the spinodal wave if the instability occurs in the uniform solution film with concentration C_c . If the concentration deviation ΔC does not involve the spinodal precipitation, the resulting concentration wave will be shown as the blue curve in Fig.(c) which is the superposition of the black curve and $\Delta C(x)$ in Fig.(b). In reality, the $\Delta C(x)$ is a part of solution concentration $C(x)$ and it will involve the spinodal precipitation. This leads an enlargement of the amplitude of the blue wavy curve, i.e., the red curve in Fig.(c). Therefore, even at the situation of existing a moderate concentration gradient caused by solvent evaporation the spinodal precipitation can still be observed.

This is described in the revised manuscript and supplementary information. (See Page 13 of Manuscript, Supplementary Note 8, Supplementary Fig. 15)

[1] Y. Shang, D. Kazmer, M. Wei, C. Barry, J. Mead, Numerical simulation of the self-assembly of a polymer–polymer–solvent ternary system on a heterogeneously functionalized substrate. *Polym. Eng. Sci.* 50, 2329-2339 (2010).

Figure. (a) Schematic illustration of solvent evaporation and solution concentrating process,

where a much higher mobility of solvent molecules in comparison with solute molecules is assumed. For a clear illustration the solvent molecules (blue circles) and the solute molecules (black circles) were drawn in different layers. (I) Thin solution film sandwiched between two plates; (II) A solvent molecule leaves from sample edge and a vacancy (numbered with 1) is created; (III) Vacancy 1 diffuses inward through replacement with a solvent molecule; (IV) Vacancy 1 moves further inward and next vacancy (numbered with 2) is created by solvent evaporation; (V) Vacancies move further inwards. Red arrow shows that a solvent molecule can move into a vacancy from various directions; (VI) Reduction of film thickness. (b) Schematic illustration of concentration deviation from sample center to edge (red line) and concentration wave in a solution film with average concentration C_C . (c) Schematic illustration of concentration waves when $\Delta C(x)$ does not involve spinodal precipitation (SP) (blue curve) and $\Delta C(x)$ involves the spinodal precipitation (red curve).

9. The impact of PS on transistor performance is unclear. Figure 4g can be better labeled.

Response: After vertical phase separation the PS is located between the C8-BTBT and SiO₂ films because the stronger interaction between the PS and SiO₂. The PS surface has less charge traps than the SiO₂ surface. Therefore the transistors made with solutions of C8-BTBT+PS mixture have larger charge mobility than the transistors without the PS. We clarified this in the revised manuscript. (See Page 13 of Manuscript)

We replaced original Fig.4g with a better image and labeled. (See Page 23 of Manuscript)

10. What are the absolute thickness of the high and low structures?

Response: AFM measurement shows that the thicknesses of the generated patterns are in the range of 40~90nm depending on the compositions and concentrations of the solutions used for pattern generation. The wave structures were formed in liquid film and subsequently deposited to the substrates. During the drying of the samples, the solute molecules tend to accumulate to the early-stage deposited structures (like center part of a line). This results clean spaces between the deposited materials, i.e., the thickness of low structures is zero. We have mentioned this in the revised manuscript. (See Page 5, Page 13 of Manuscript)

REVIEWER COMMENTS

Reviewer #1 (Remarks to the Author):

See Attachment

I appreciated the authors that they have kindly addressed almost all my comments in the revised manuscript. However, I still have some problems about this manuscript. I will recommend to accept the manuscript if authors can further address these problems.

1. In Page 14, the authors say that there are pinholes in the PS layer, can they provide any solid evidence?

2. In Page 14, the authors demonstrate that a PS layer is emerged between the C₈-BTBT and SiO₂ induced by interacting of SiO₂ with the PS. However, phase separation between C₈-BTBT and PS may be caused by the difference in the surface energies other than the interaction of SiO₂ with the PS.

3. In Page 14, the authors explain that the device performance is largely limited by the traps on SiO₂ substrate. However, the poor device performance may be mainly due to large contact resistance and low crystal quality, can they improve the device performance from improving crystallinity and the contact between C₈-BTBT and electrodes?

4. There should be more electrical data in prepared OFETs, such as statistical distribution of mobility, on/off current ratio, and threshold voltage.

5. How about the hysteresis in the forward and reverse transfer characteristic curves of the device under a drain bias?

6. There should be more information in prepared P₃HT and perylene patterns. Such as, AFM and XRD.

7. For evaluating mobility, how about the per unit area of capacitance for the hybrid dielectrics of PS/SiO₂.

8. Many literatures used PDMS and silicon templates to fabricate organic semiconductor patterns, and many reports obtained single crystal arrays. There should be a table to compare this work and literatures for clarifying advantages and disadvantages to guide preparation of organic semiconductor patterns and other nanomaterials.

Reviewer #2 (Remarks to the Author):

The authors have sufficiently answered all reviewer points, the manuscript is suitable for publication.

Response to Reviewer's Comments

We thank reviewer for the valuable comments. We have carried out necessary experiments, and the manuscript has been revised as suggested by the reviewer. Please see the point-by-point response to the comments listed below.

Q1. In Page 14, the authors say that there are pinholes in the PS layer, can they provide any solid evidence

Response: We attempted use “pinholes” to explain the obtained low charge mobility, and this speculation was from literatures (for instance: Haase et al. *Adv. Electron. Mater.* **4**, 1800076 (2018)) where the importance of PS-coverage on substrate was stated. There was no direct evidence of pinholes from our experiments.

Inspired by the comment of reviewer, we have fabricated new devices and reconsidered this problem with more analysis (see response to Q3 below). We agree with the reviewer that the charge mobility of our devices is mainly affected by the contact resistance and crystal quality of the semiconductor.

We have corrected this point in the revised manuscript (Page 14).

Q2. In Page 14, the authors demonstrate that a PS layer is emerged between the C8-BTBT and SiO₂ induced by interacting of SiO₂ with the PS. However, phase separation between C8-BTBT and PS may be caused by the difference in the surface energies other than the interaction of SiO₂ with the PS.

Response: We did more literature search (Liu et al. *Organ. Electron.* **12**, 1446 (2011). Yuan et al. *Nat. Commun.* **5**, 3005 (2014). Arias et al. *Adv. Mater.* **18**, 2900 (2006)) and agree with the reviewer that the surface energy difference is more proper explanation.

During phase separation, PS layer is emerged between the C8-BTBT and SiO₂ which is caused by the difference in the surface energies. As the methyl-terminated C8-BTBT has lower surface energy than either substrate or PS surfaces, the energy can be minimized when PS segregates to the substrate and C8-BTBT surface is exposed.

We clarified this in the revised manuscript (Page 14), and references are provided.

Q3. In Page 14, the authors explain that the device performance is largely limited by the traps on SiO₂ substrate. However, the poor device performance may be mainly due to large contact resistance and low crystal quality, can they improve the device performance from improving crystallinity and the contact between C8-BTBT and electrodes?

Response: We have fabricated new devices and measured them. 5nm MoO₃ layer was

introduced between the semiconductor and gold electrodes to reduce the contact resistance. The device performance has been improved (see Fig.Q3-a, Fig.Q3-b below). The obtained charge mobility is $2.5 \text{ cm}^2 \cdot \text{V}^{-1} \cdot \text{S}^{-1}$ and on/off current ratio is $\sim 10^6$. As suggested from the reviewer, we investigated the hysteresis behavior of the devices (Fig.Q3-c below). The moderate hysteresis suggests that the charge trapping effect is less significant, as the hysteresis obtained with forward and reverse gate-voltage scans reflects charge trapping process. Therefore, the charge mobility is mainly limited by contact resistance and low crystal quality of the semiconductor. Although a MoO_3 layer is introduced to reduce the contact resistance, the charge injection can still be improved which is proved from the sublinear characteristic of the output curves at low drain voltage (see Fig.Q3-a below). During device fabrication, only moderate thermal annealing (at 60°C) in N_2 atmosphere was processed. Solvent vapor annealing of the patterned lines (to improve quality of crystallization) was failed which caused break of lines into multi-segments due to material aggregation.

We clarified this in the revised manuscript (Page13-14, Fig.4e and Fig.4f) and supplementary information (Supplementary Fig.17a).

Fig.Q3. Output (a) and transfer (b) curves taken from a FET device made with C8-BTBT+PS wires. (c) Forward and reverse transfer characteristic curves recorded from the same device.

Q4. There should be more electrical data in prepared OFETs, such as statistical distribution of mobility, on/off current ratio, and threshold voltage.

Response: For the newly fabricated devices the obtained charge mobility is in the range of 1.7~2.5 $\text{cm}^2\cdot\text{V}^{-1}\cdot\text{S}^{-1}$, on/off current ratio varies between 1.4×10^6 and 7.5×10^6 , threshold voltage (V_{th}) varies between 0V and +3V.

We have mentioned this in the revised manuscript (Page 14).

Q5. How about the hysteresis in the forward and reverse transfer characteristic curves of the device under a drain bias?

Response: The hysteresis of the devices is not large. The threshold voltage shift (ΔV_{th}) between forward and reverse gate-voltage scans is less than 3V (Please see the Fig.Q3-c).

We have described this in the revised manuscript (Page 14), and the hysteresis behavior is shown in the revised Supplementary Fig.17a.

Q6. There should be more information in prepared P3HT and perylene patterns. Such as, AFM and XRD.

Response: Follow the suggestion of the reviewer we prepared new P3HT and perylene patterns and carried out AFM measurement and XRD analysis. Solutions of P3HT (regioregular) and perylene were prepared with concentration 10mg/ml in chlorobenzene. The pattern generation process is identical to that for C8-BTBT patterning. For the P3HT, array of lines is preferred pattern generated and the pattern morphology is sensitive to small variation of experimental condition. Modified line patterns, such as modulated lines or mosaic-like patterns were observed. No crystal grains were observed for the P3HT structure with the AFM investigation which is anticipated for polymer materials (see Fig.Q6-1a and Fig.Q6-1b below). The ~2nm sized feature in the Fig.Q6-1b might be originated from polymer chain-staking or chain-folding. X-ray analysis shows that the material is mainly amorphous with a little sign of crystallization which is probably caused by chain-staking or folding of the P3HT (Fig.Q6-2a) [1]. This agrees with the observed nanofeature in Fig.Q6-1b. For the perylene, line patterns were found. Both AFM and X-ray analysis showed strong crystalline structure/phase (see Fig.Q6-1c, Fig.Q6-1d, Fig.Q6-2b below).

The patterned perylene crystal is very different from the patterned C8-BTBT. Although both C8-BTBT and perylene are small molecule semiconductor materials, the patterned perylene shows single crystal morphology. This might be originated from different mechanism of crystallization of the two materials. Crystallization is controlled by nucleation and growth process. For the C8-BTBT, the crystallization is probably mainly controlled by nucleation process (number of nuclei is large, the growth of crystal is slow). Therefore, solvent vapor annealing (SVA) is often used to generate proper sized grains after C8-BTBT film deposition [2]. For the perylene, the crystallization process could be dominated by growth process, and large sized perylene crystals can be

prepared by simple solution drop-casting [3].

Although patterned perylene crystals can be fabricated here, the charge mobility of perylene crystal is low ($10^{-2}\sim 10^{-4}$ $\text{cm}^2\cdot\text{V}^{-1}\cdot\text{S}^{-1}$) [3,4]. Therefore, finding organic semiconductor materials to generate patterned single crystals with high charge mobility is interesting for device fabrication.

We described this in the revised manuscript (Page 11), and detailed in supplementary information (Supplementary Note 6, Supplementary Fig. 10 and Fig. 11). References are given in the supplementary information.

[1] Zhokhavets, U., Erb, T., Hoppe, H., Gobsch, G., Serdar Sariciftci, N. Effect of annealing of poly(3-hexylthiophene)/fullerene bulk heterojunction composites on structural and optical properties. *Thin Solid Films* **496**, 679-682 (2006).

[2] Liu, C., Minari, T., Lu, X., Kumatani, A., Takimiya, K., Tsukagoshi, K. Solution-processable organic single crystals with bandlike transport in field-effect transistors. *Adv. Mater.* **23**, 523-526 (2011).

[3] Liao, Q., Zhang, H., Zhu, W., Hua, K., Fu, H. Perylene crystals: tuning optoelectronic properties by dimensional-controlled synthesis. *J. Mater. Chem. C* **2**, 9695 (2014).

[4] Lee, J-W., Kang, H-S., Kim, M-K., Kim, K., Cho, M-Y., Kwon, Y-W., Joo, J. Electrical characteristics of organic perylene single-crystal-based field-effect transistors. *J. Appl. Phys.* **102**, 124104 (2007).

Fig. Q6-1. AFM images of patterned P3HT and Perylene with different magnifications. Image of patterned P3HT (a) and zoom-in image of local area with a line (b); Patterned perylene lines

(c) and zoom-in image of local area on the top of a line.

Fig. Q6-2. X-ray diffraction spectra taken from patterned P3HT (a) and perylene (b) samples. The inset of (a) shows a schematic illustration of chain-stacking of P3HT.

Q7. For evaluating mobility, how about the per unit area of capacitance for the hybrid dielectrics of PS/SiO₂.

Response: The capacitance of unit area C_0 is expressed as:

$$C_0 = \frac{\epsilon_0 \epsilon_{ps} \epsilon_{ox}}{\epsilon_{ps} d_{ps} + \epsilon_{ox} d_{ox}}$$

Where ϵ_{ps} and ϵ_{ox} is the relative permittivity of PS and SiO₂ layer, respectively; d_{ps} and d_{ox} is the thickness of PS and SiO₂ layer; ϵ_0 is vacuum permittivity. Taking $\epsilon_{ps} \sim 2.5$, $\epsilon_{ox} \sim 3.9$, $d_{ps} \sim 30\text{nm}$, $d_{ox} \sim 300\text{nm}$, $\epsilon_0 \sim 8.854 \times 10^{-12} \text{ F/m}$, we have $C_0 \sim 6.93 \times 10^{-5} \text{ (F/m}^2\text{)}$.

This is described in the revised manuscript (Page 14).

Q8. Many literatures used PDMS and silicon templates to fabricate organic semiconductor patterns, and many reports obtained single crystal arrays. There should be a table to compare this work and literatures for clarifying advantages and disadvantages to guide preparation of organic semiconductor patterns and other nanomaterials.

Response: In the revised manuscript we described the difference between this work and some other stamp-based patterning methods. A table is provided in the revised supplementary information, where 15 fabrication techniques for patterning organic semiconductors (OSC) are listed (see the table below). Following description has been

added to the revised manuscript (Page 15):

“In this work we used PDMS stamps to confine solution and investigate pattern formation of organic semiconductors (OSC). Comparing with other works of OSC patterning using stamp techniques (Supplementary Table 1) this work has its own distinctiveness: (i) In this work, the structure on a PDMS stamp is used to control solution thickness only, and the lateral dimension of the structure is not important. In contrast, for the methods listed in the table, the lateral dimension of the structures on stamps is crucial to define the size and geometry of patterned OSC. (ii) The patterns formed in this work are self-organized, instead of artificially designed. (iii) In this work, the patterns are formed under non-equilibrium condition, which is different from those where the patterned materials are formed under (or close to) equilibrium condition. On the other hand, this work also shares some common features with many other works, such as surface energy tuning of stamps/substrates, flexibility control of stamps for firm contact during printing etc. With further development this work may have potential to fabricate nanomaterials with low cost. The challenge is that the variety of formed patterns are limited, and the size of patterned structures needs to be controlled very carefully.”

We described this in the revised manuscript (Page 15) and Supplementary Table 1.

(Please see the Table 1 in next few pages)

Table 1. Examples of patterning organic semiconductor (OSC) with stamping techniques.

No	Schematic illustrations	Brief description	Ref
1	 (a) Solvent, OSC, Substrate (b)	Patterned PDMS stamp is dipped into solvent to absorb the solvent molecules. Subsequently, the PDMS stamp is firmly contacted with an organic semiconductor (OSC) film. The film is selectively dissolved (with solvent diffused out) and patterned.	[1]
2	 (a) OSC, Substrate (b)	Patterned PDMS stamp is dipped into OSC solution and dried afterwards. Then, the PDMS stamp is contacted onto a substrate surface and the OSC molecules are transferred to the substrate surface via out-diffusion.	[2]
3	 (a) PDMS, OSC, Substrate (b)	Use a PDMS stamp to selectively remove OSC film deposited on a substrate. The PDMS stamp is firmly contacted with the precoated OSC film and the OSC molecules diffuse into the PDMS material.	[3-5]
4	 (a) PDMS, SAM, Substrate (b) OSC	Selective growth of vertical nanowires of OSC on patterned substrate made with stamping technique. The fabrication principle used is that the nanowires, grown by physical vapor deposition, tend to grow within the hydrophilic areas.	[6]
5	 (a) OSC, SAM, PDMS (b) Substrate	Micro/nano molding. Micro/nano structured material is produced by selectively inking a PDMS stamp with OCS solution. Then, the structured OCS templated by the PDMS is transferred to a substrate surface.	[7]

6	 (a) PDMS stamp with a patterned surface. (b) The stamp is pressed against a substrate, transferring an OSC film to the substrate surface.	Material transfer from PDMS stamp to substrate. OSC film is formed on the structured surface of a PDMS stamp by spin-coating or other deposition techniques. Then, the OSC is selectively transferred onto a substrate surface.	[8]
7	 (a) Solution is dispensed onto a substrate and a patterned PDMS is brought to contact with the inked substrate. The solution is segregated into the grooves of the stamp through capillary effect and dried there. (b) The stamp is pressed against the substrate, picking up the solution from the grooves.	Solution is dispensed onto a substrate and a patterned PDMS is brought to contact with the inked substrate. The solution is segregated into the grooves of the stamp through capillary effect and dried there.	[9,10]
8	 (a) Vapor growth of OSC single crystal arrays on a patterned substrate through controlled nucleation. The nucleation sites were defined by soft contact printing with a PDMS stamp. (b) OSC Crystals are formed on the substrate surface.	Vapor growth of OSC single crystal arrays on a patterned substrate through controlled nucleation. The nucleation sites were defined by soft contact printing with a PDMS stamp.	[11]
9	 (a) Use a PDMS stamp to print surface energy pattern on a substrate, and the deposited OSC solution is patterned with the wetting contrast on the substrate and selectively dried on the hydrophilic areas. (b) OSC is deposited on the substrate surface.	Use a PDMS stamp to print surface energy pattern on a substrate, and the deposited OSC solution is patterned with the wetting contrast on the substrate and selectively dried on the hydrophilic areas.	[12]
10	 (a) Fabrication of OSC arrays by a stamp with an inclined slope structure. Solution under such a stamp is distributed under the inclined features caused by capillary effect during drying. (b) OSC is deposited on the substrate surface.	Fabrication of OSC arrays by a stamp with an inclined slope structure. Solution under such a stamp is distributed under the inclined features caused by capillary effect during drying.	[13,14]

11	 (a) PDMS stamp on substrate with solution and solute. (b) OSC wires formed at channel edges.	Stamp guided side deposition. Drying solution confined in channels of structured PDMS stamp induces a convection which transfer OSC molecules to the channel edges. Submicron sized OSC wires can be produced.	[15,16]
12	 (a) Liquid suspension, PDMS stamp, Porous AAO, Vacuum. (b) OSC wires transferred to substrate.	Dispersed OSC wires in liquid suspension are deposited into the trenches of a stamp forced by solution outflow through porous materials induced by a pressure difference. Then, the aligned patterned wires are transferred onto substrate surface.	[17]
13	 (a) OSC material at channel entrance. (b) OSC material filling channels by capillary action.	An OSC material with low melting point is placed at the entrance of channels of a PDMS stamp. Heating the OSC to a temperature above its melting point and the channels are filled with capillary action.	[18]
14	 (a) PDMS stamp on Au stripes. (b) OSC wires formed in the space underneath the stamp.	A structured stamp is placed on pre-patterned electrode stripes. Solution dispensed at the edge of the stamp is sucked into underneath space of the stamp with capillary force and dried there.	[19,20]
15	 (a) Stamp guiding crystal nucleation and growth of OSC materials. (b) OSC pattern transferred to substrate.	Stamp is used to guide the crystal nucleation and growth of OSC materials. OSC vapor is deposited and crystallized on the edges of pillar structure with the help of wetting contrast between top and vertical areas induced by surface roughness change. The OSC pattern can be transferred to a substrate.	[21]

- [1] Kim, K., Jang, M., Lee, M., An, T. K., Anthony, J. E., Kim, S. H., Yang, H., Park, C. E. Unified film patterning and annealing of an organic semiconductor with micro-grooved wet stamps. *J. Mater. Chem. C* **4**, 6996-7003 (2016).
- [2] Lee, K., Kim, J., Shin, K., Kim, Y. S. Micropatterned crystalline organic semiconductors via direct pattern transfer printing with PDMS stamp. *J. Mater. Chem.* **22**, 22763 (2012).
- [3] Dickey, K. C., Subramanian, S., Anthony, J. E., Han, L., Chen, S., Loo, Y. Large-area patterning of a solution-processable organic semiconductor to reduce parasitic leakage and off currents in thin-film transistors. *Appl. Phys. Lett.* **90**, 244103 (2007).
- [4] Park, H-L., Lee, B-Y., Kim, S-U., Suh, J-H., Kim, M-H., Lee, S-D. Importance of surface modification of a microcontact stamp for pattern fidelity of soluble organic semiconductors. *J. Micro/Nanolith. MEMS MOEMS* **15**, 013501 (2016).
- [5] Bae, I., Kang, S. J., Shin, Y. J., Park, Y. J., Kim, R. H., Mathevet, F., Park, C. Tailored single crystals of triisopropylsilylethynyl pentacene by selective contact evaporation printing. *Adv. Mater.* **23**, 3398–3402 (2011).
- [6] Zhao, Y. S., Zhan, P., Kim, J., Sun, C., Huang, J. Patterned growth of vertically aligned organic nanowire waveguide arrays. *ACS Nano* **3**, 1630-1636 (2010).
- [7] Park, K. S., Cho, B., Baek, J., Hwang, J. K., Lee, H., Sung, M. M. Single-crystal organic nanowire electronics by direct printing from molecular solutions. *Adv. Funct. Mater.* **23**, 4776–4784 (2013).
- [8] Takakuwa, A., Azumi, R. Influence of solvents in micropatterning of semiconductors by microcontact printing and application to thin-film transistor devices. *Jpn. J. Appl. Phys.* **47**, 1115-1118 (2008).
- [9] Jo, P. S., Vailionis, A., Park, Y. M., Salleo, A. Scalable fabrication of strongly textured organic semiconductor micropatterns by capillary force lithography. *Adv. Mater.* **24**, 3269–3274 (2012).
- [10] Watanabe, S., Fujita, T., Ribierre, J., Takaishi, K., Muto, T., Adachi, C., Uchiyama, M., Aoyama, T., Matsumoto, M. Microcrystallization of a solution-processable organic semiconductor in capillaries for high-performance ambipolar field-effect transistors. *ACS Appl Mater Interfaces* **8**, 17574-17582 (2016).
- [11] Briseno, A. L., Mannsfeld, S. C. B., Ling, M. M., Liu, S., Tseng, R. J., Reese, C., Roberts, M. E., Yang, Y., Wudl, F., Bao, Z. Patterning organic single-crystal transistor arrays. *Nature* **444**, 913-917 (2006).
- [12] Briseno, A. L., Roberts, M., Ling, M. M., Moon, H., Nemanick, E. J., Bao, Z. Patterning organic semiconductors using “dry” poly(dimethylsiloxane) elastomeric stamps for thin film transistors. *J. Am. Chem. Soc.* **128**, 3880-3881(2006).
- [13] Nakayama, K., Hirose, Y., Soeda, J., Yoshizumi, M., Uemura, T., Uno, M., Li, W., Kang, M. J., Yamagishi, M., Okada, Y., Miyazaki, E., Nakazawa, Y., Nakao, A., Takimiya, K., Takeya,

- J. Patternable solution-crystallized organic transistors with high charge carrier mobility. *Adv. Mater.* **23**, 1626–1629 (2011).
- [14] Diao, Y., Shaw, L., Bao, Z., Mannsfeld, S. C. B. Morphology control strategies for solution processed organic semiconductor thin films. *Energy Environ. Sci.* **7**, 2145–2159 (2014).
- [15] Li, J., Chang, X., Li, S., Shrestha, P. K., Tan, E. K. W., Chu, D. High-resolution electrochemical transistors defined by mould-guided drying of PEDOT:PSS liquid suspension. *ACS Appl. Electron. Mater.* **2**, 2611(2020).
- [16] Li, S., Chun, Y. T., Zhao, S., Ahn, H., Ahn, D., Sohn, J. I., Xu, Y., Shrestha, P., Pivnenko, M., Chu, D. High-resolution patterning of solution-processable materials via externally engineered pinning of capillary bridges. *Nat. Commun.* **9**, 393 (2018).
- [17] Oh, J. H., Lee, H. W., Mannsfeld, S., Stoltenberg, R. M., Jung, E., Jin, Y. W., Kim, J. M., Yoo, J. B., Bao, Z. Solution-processed, high-performance n-channel organic microwire transistors. *PNAS* **106**, 6065-6070 (2009).
- [18] Kim, A., Jang, K-S., Kim, J., Won, J. C., Yi, M. H., Kim, H., Yoon, D. K., Shin, T. J., Lee, M-H., Ka, J-W., Kim, Y. H. Solvent-free directed patterning of a highly ordered liquid crystalline organic semiconductor via template assisted self-assembly for organic transistors. *Adv. Mater.* **25**, 6219–6225 (2013).
- [19] Cavallini, M., D’Angelo, P., Criado, V. V., Gentili, D., Shehu, A., Leonardi, F., Milita, S., Liscio, F., Biscarini, F. Ambipolar multi-stripe organic field-effect transistors. *Adv. Mater.* **23**, 5091–5097 (2011).
- [20] Zhang, X. J., Jie, J., Deng, W., Shang, Q., Wang, J., Wang, H., Chen, X., Zhang, X. H. Alignment and patterning of ordered small-molecule organic semiconductor micro-/nanocrystals for device applications. *Adv. Mater.* **28**, 2475–2503 (2016).
- [21] Wu, Y., Feng, J., Jiang, X., Zhang, Z., Wang, X., Su, B., Jiang, L. Positioning and joining of organic single-crystalline wires. *Nat. Commun.* **6**, 6737 (2015).

REVIEWERS' COMMENTS

Reviewer #1 (Remarks to the Author):

The reviewer read the response letter and revised manuscript, and believe the authors have addressed the reviewer's concerns, and would like to recommend the acceptance of the manuscript.